# ^18^F-fluorodeoxyglucose (^18^F-FDG) Functionalized Gold Nanoparticles (GNPs) for Plasmonic Photothermal Ablation of Cancer: A Review

**DOI:** 10.3390/pharmaceutics15020319

**Published:** 2023-01-18

**Authors:** Mariano Pontico, Miriam Conte, Francesca Petronella, Viviana Frantellizzi, Maria Silvia De Feo, Dario Di Luzio, Roberto Pani, Giuseppe De Vincentis, Luciano De Sio

**Affiliations:** 1Department of Radiological Sciences, Oncology and Anatomo-Pathology, Sapienza, University of Rome, 00185 Rome, Italy; 2Institute of Crystallography CNR-IC, National Research Council of Italy, Monterotondo, 00015 Rome, Italy; 3Department of Medico-Surgical Sciences and Biotechnologies, Research Center for Biophotonics, Sapienza University of Rome, 04100 Latina, Italy

**Keywords:** ^18^F-FDG, gold nanoparticles, plasmonic photothermal ablation

## Abstract

The meeting and merging between innovative nanotechnological systems, such as nanoparticles, and the persistent need to outperform diagnostic-therapeutic approaches to fighting cancer are revolutionizing the medical research scenario, leading us into the world of nanomedicine. Photothermal therapy (PTT) is a non-invasive thermo-ablative treatment in which cellular hyperthermia is generated through the interaction of near-infrared light with light-to-heat converter entities, such as gold nanoparticles (GNPs). GNPs have great potential to improve recovery time, cure complexity, and time spent on the treatment of specific types of cancer. The development of gold nanostructures for photothermal efficacy and target selectivity ensures effective and deep tissue-penetrating PTT with fewer worries about adverse effects from nonspecific distributions. Regardless of the thriving research recorded in the last decade regarding the multiple biomedical applications of nanoparticles and, in particular, their conjugation with drugs, few works have been completed regarding the possibility of combining GNPs with the cancer-targeted pharmaceutical fluorodeoxyglucose (FDG). This review aims to provide an actual scenario on the application of functionalized GNP-mediated PTT for cancer ablation purposes, regarding the opportunity given by the ^18^F-fluorodeoxyglucose (^18^F-FDG) functionalization.

## 1. Introduction

Despite many years of research and high financial costs, definitive treatment for cancer remains currently beyond our reach. According to the World Health Organization report, cancer caused the death of 9.6 million people and over 18 million new cancer cases in 2018, ranking as the second cause of death worldwide [1]. Although a vast number of cancer treatments, most notably radiotherapy and chemotherapy, have been developed, these are not without flaws, and patients continue to suffer from side effects, most remarkably due to the highly toxic nature of the chemotherapeutic agents. Furthermore, some types of cancer appear to remain incurable with current radiotherapy and chemotherapy approaches. Because of their potential for early detection, accurate diagnostics, and the possibility for personalized treatment, interest in the application of nanoparticles for in vivo diagnostic and therapeutic purposes has grown over the last decade [2]. Nanotechnology, which has significantly advanced over the last several decades, now provides a new targeted cancer treatment with less harm to surrounding normal cells [3,4]. Because nanoparticles frequently lack pathology-targeting mechanisms, various functional biomolecules have been developed to bind with nanoparticles and make them functional [5]. Many studies have demonstrated that functional nanoparticles are more effective in cancer treatment and cellular targeting, highlighting how gold nanoparticles (GNPs) are very promising candidates for various biomedical applications in cancer diagnostics and therapeutics [6,7,8]. Despite the flourishing research recorded in the last decade regarding the multiple biomedical applications of nanoparticles and, in particular, their conjugation with drugs, the possibility of combining gold nanoparticles with the cancer-targeted molecule fluorodeoxyglucose (FDG) has been less explored. The opportunity to combine effective and specific molecular targeting with cancer-targeted hyperthermia can constitute a time of growth in the process of seeking new, more effective and safer strategies for ablative cancer treatment. This review article aims to provide a summary of the actual applications of functionalized GNP-mediated photothermal therapy (PTT) for cancer ablation purposes. Particular regard is reserved for the analysis of the available gold nanoparticles’ functionalization methods with ^18^F-fluorodeoxyglucose (^18^F-FDG).

## 2. Current Limitations of Conventional Cancer Therapies

The heterogeneous nature of cancer has so far prevented a fully comprehensive approach to effective treatment. Although providing some therapeutic efficacy, conventional approaches are limited by their risk to healthy cells, their potential to destroy the immune system, or by conferring an increased risk for the development of secondary cancer. Conventional treatments have a variety of side effects that cause both physical and psychological stress in patients. Chemotherapeutics produce a wide range of toxicities, including hematotoxicity, cardiotoxicity, gastrointestinal toxicity, neurotoxicity, and nephrotoxicity [9]. These drugs target multiple cells, making it difficult to distinguish between cancerous and normal cells. The intrinsic non-specificity of these drugs restricts their maximum allowable dose. In addition, chemotherapeutics are rapidly eliminated from the body, mostly through renal clearance. Therefore, administration of a high dosage is needed to avoid rapid elimination and for a broader distribution of the drug to the targeted tissues. Ionizing radiation therapy, along with chemotherapy, is a key cancer treatment technique used by more than half of all cancer patients [10,11,12]. Despite advances in medical imaging and radiation sources up to image-guided radiotherapy, limiting the curative dose of radiation to malignant cells while sparing neighboring healthy tissues remains a significant problem [13,14]. Non-ionizing radiation sources, such as microwaves, radiofrequency, and ultrasound, are also used to treat cancer by producing heat. PTT or microwave therapy can destroy malignant cells by producing hyperthermia through the internalization of therapeutic drugs with a high photothermal conversion efficiency when exposed to external laser irradiation. PTT has recently gained a great deal of attention because of its manageable treatment approach, high tumor eradication efficacy, and low adverse effects on healthy cells [6]. Hyperthermia is a medical treatment that involves exposing tissues to slightly greater temperatures than normal in order to attack and eradicate cancer cells. Tumor tissues have an unusually chaotic and sparse circulatory architecture, which makes heat dissipation difficult and makes them more vulnerable to hyperthermia than normal tissue. Cancer cells can be selectively eradicated at temperatures ranging from 40 to 44 °C, when detrimental consequences such as DNA damage, protein denaturation, and cellular membrane disruption occur, resulting in tumor tissue ablation. Traditional hyperthermia treatments administer heat to patients using hot water, microwaves, ultrasound, and radiofrequency radiation. While these approaches heat tumor tissues, they frequently lack selectivity toward healthy tissues [15,16].

## 3. Gold Nanoparticles for PTT

The scientific literature provides a plethora of valuable approaches for the synthesis of gold nanoparticles with extremely high control over their size, shape, and physiochemical properties. Such a high control can be regarded as the key that opens enormous applicative opportunities since the optoelectronic, catalytic, physical, and textural properties of gold nanoparticles are morphology dependent. The most relevant property of gold nanoparticles, which makes them appealing for biomedical applications, is related to the Localize Surface Plasmon Resonance (LSPR) phenomenon. This is essentially due to the collective oscillation of surface electrons occurring when gold nanoparticles absorb light at a suitable wavelength (called plasmon resonance wavelength), thus resulting in the appearance of an intense peak in their absorption spectrum (plasmon band) [17]. Provided that LSPR conditions occur, light absorption is followed by radiative and non-radiative relaxation processes. The main non-radiative relaxation process is the generation of heat (plasmonic heating), which enables the confining of an extremely high amount of heat at the nanoscale [4]. For this reason, gold nanoparticles can be used as heat nano sources in PTT [18]. To be more precise, when PTT is realized by plasmonic gold nanoparticles, Plasmonic Photothermal Therapy (PPTT) is defined. The efficiency of plasmonic heating is strictly related to the plasmonic properties of the investigated gold nanoparticles, such as the wavelength and intensity of the plasmon band. The position of the plasmon band is strongly affected by the morphology and surface chemistry of gold nanoparticles and can be tuned with suitable synthesis methods and a fully comprehensive approach to post-synthesis functionalization. The synthesis approaches of gold nanoparticles can be categorized into two groups: top-down and bottom-up techniques, respectively. The former includes physical approaches, such as lithography, photochemical, electrochemical, templating, sonochemical, and thermal reduction techniques that rely on the ablation of bulk materials up to reach nanosized structures [19]. Conversely, in bottom-up approaches, gold nanoparticles are synthesized from a suitable inorganic precursor by assembling the nanostructure atom by atom [20]. Among the bottom-up approaches, colloidal synthesis methods are generally preferred because they are affordable, do not require specific and expensive laboratory equipment, and allow for finely controlled size shape and surface chemistry of the resulting nanostructure, according to the desired chemical-physical properties. Besides achieving colloidal gold nanoparticle dispersions with a high degree of monodispersity, colloidal synthesis allows the preparation of nanoparticles with peculiar surface chemistry, characterized by surface functional groups that enable colloidal stability and, remarkably, can be replaced and/or engineered with different functional molecules suitable for the specific application. The Turkevich method and the Brust method are among the most diffused techniques for the colloidal synthesis of spherical gold nanoparticles in a dimensional regime ranging from 5 to 150 nm [19,21]. However, to improve monodispersity, “seed-mediate” techniques are preferred. Seed-mediate techniques are essentially two-step processes, consisting of a first stage devoted to the preparation of Au nuclei (seed) and a second stage where the pre-synthesized nuclei grow in extremely controlled conditions, suited to achieve a uniform size distribution [22]. The reliable medical applications of GNPs in PTT require biocompatible nanostructures able to absorb light in the “biological window,” namely in a wavelength range from 700 nm to 900 nm where biological tissue absorption is low and, thus, the penetration depth of the radiation is high [23]. Such a condition enables, in principle, to ablate deeper tumor sites without affecting the surrounding healthy tissues [4]. This requirement is fulfilled by anisotropic gold nanoparticles such as gold nanorods and gold bipyramids or complex gold nanostructures including nanocubes, nanocages, nanoshells, and nanostars [24] whose main properties will be briefly overviewed in the following section.

### 3.1. Gold Nanorods

Gold nanorods (GNRs) are among the most utilized gold-based nanostructures, as shown by the considerable development of photothermal strategies based on the GNRs [6]. GNRs are commonly accepted as suitable PTT agents due to their good cell permeability and LSPR properties. Indeed, GNRs exhibit two plasmon resonance peaks: the transverse band and the longitudinal band, respectively [2]. The transverse band is typically centered at 520 nm, while the longitudinal band can be tuned from visible wavelengths to near-infrared (NIR) wavelengths according to the aspect ratio of the GNRs that can be selected a priori in the synthesis step. As the aspect ratio increases, the longitudinal peak shifts toward the NIR region of the absorption spectrum (red-shift) [15]. The most investigated approach for the synthesis of GNRs is a seed-mediated method. It consists of the production of gold seeds by reducing a gold precursor with a strong reducing agent, such as sodium borohydride. The gold seeds are then introduced, a solution of the metal precursor containing a weak reducing agent (e.g., ascorbic acid), a structure-directing agent, and suitable additives to prevent further nucleation and to promote the anisotropic growth of gold-based nanostructures. The dimensions of the obtained GNRs can be controlled by playing on the ratio among the different employed reactants [25]. The longitudinal plasmon band of GNRs shows an intense absorption peak in the NIR wavelength range; accordingly, GNRs are optimal nanoparticles for performing PTT. Indeed, GNRs exhibit a high photothermal efficiency under NIR light that possesses a penetration depth of about 2.5–3 mm; therefore, they are suitable light-to-heat optical transducers for tumor ablation, as will be described in the next sections.

### 3.2. Gold Nanoshells

Gold nanoshells (GNSs) are spherical nanostructures characterized by dielectric and thin gold layers that alternate, resulting in a complex core-shell dielectric/metal nanostructure. GNSs exhibit peculiar LSPR properties resulting from the coupling strength and energy difference between the plasmon band of the inner shells and the plasmon band of the outer shells. The LSPR wavelength of gold nanoshells can be tuned from visible to NIR by adjusting the dimensional ratio of the core radius and shell thickness. The synthesis of GNSs, although more challenging if compared to the synthesis of gold nanorods, does not require cytotoxic reactants such as CTAB; therefore, GNSs are suitable for biomedical applications such as photothermal therapy. Interesting is the research of Loo et al. [26] and Fekrazad et al. [27] which demonstrated that the modification of GNSs with anti-Human Epidermal Growth Factor Receptor (HER)-2 antibodies such as Herceptin^®^ (Genentech, South San Francisco, CA, USA) allows selective GNS accumulation in breast cancer cells.

### 3.3. Gold Nanocages

Gold nanocages (GNCs) are a novel class of nanostructures recently developed by Xia et al. [28]. GNCs have a hollow structure with porous walls. By varying the quantity of the gold precursor, the position of the plasmonic signal of the GNCs can be tuned in the range from 600 nm to 1200 nm. The possibility of tuning the surface plasmon resonance position, hollow interiors, and porous walls makes GNCs suitable for PTT and drug delivery [29].

### 3.4. Other Novel Nanostructures

Further novel nanostructures, gold nanostars (GNSTs) and gold nanopopcorns (GNPCs), are made by an internal core from which spiking extremities point outward, thus providing anisotropy to the whole nanostructure. The size of these extremities drives the optical properties of GNSTs and GNPCs. In particular, the interplay between the plasmons and the plasmons of the core affects the absorption intensity in the NIR wavelength region [30].

## 4. GNP Engineering for Active Tumor Targeting

One of the main pursuits in achieving a reliable cancer-targeted nanoplatform is to modify the surface of GNPs with biomolecules that are able to ensure biocompatibility and selectively recognize cancer cells [3]. Beyond that, a suitable post-synthesis functionalization procedure is needed to address some shortcomings of GNPs, such as non-specific biodistribution, limited biocompatibility, rapid blood clearance, and scarce colloidal stability in physiological conditions [31]. The molecules selected for post-synthesis functionalization can enable the passive or active targeting of GNPs to specific cancer cells. Passive targeting is possible because of the enhanced permeability and retention effect (EPR). The EPR effect allows gold nanoparticles to load into tumor tissue by benefiting from the permeability of tumor vessels and retention in the tumor bed because of defective lymphatic drainage [32,33]. Because of the leaky nature of immature tumor vasculature, gold nanoparticles can passively accumulate at tumor sites, where they are likely taken into cells via non-specific receptor-mediated endocytosis [34]. Indeed, cancer cells have a higher capacity for endocytosis due to their high metabolic activity compared to normal cells [35]. GNP endocytosis is mainly accomplished through the phagocyte mechanism and non-phagocytic mechanisms (pinocytosis). Even though the EPR effect has been widely exploited to deliver GNPs, recent studies suggest that passive targeting is not as efficient as expected due to inter- and intra-tumoral heterogeneity. This affects the neo-vasculature architecture and the tumor microenvironment [36]. GNPs without targeting agents on their surface accumulate in cancer cells, as well as in healthy cells, thus increasing the possibility of causing unwanted effects. Therefore, to achieve effective tumor ablation by PTT, GNPs must be engineered to differentiate between tumoral and healthy tissues. Accordingly, the functionalization of GNPs with suitable targeted molecules is mandatory to obtain a safe, selective, and effective treatment, to decrease, as much as possible, the power density of the laser, and to minimize the probability of size effects. GNPs are prone to functionalization with several biomolecules, including antibodies, peptides, and nucleic acid aptamers, mainly through the thiol (-SH) and amine (-NH_2_) groups to identify and connect gold nanoparticles to specific target cells. The ligands selected for active targeting should possess good biocompatibility after binding to the GNP surface and preserving their biological activity [37]. Furthermore, in comparison to single functionalized GNPs, multifunctionalized GNPs have been proven to be better theragnostic agents capable of multimodal imaging diagnosis as well as targeted cancer therapy [38,39]. To even have a chance to reach their target, it has been diffusely reported that GNPs must be concealed from the host’s immune system to avoid their detection and destruction [40,41]. For this purpose, small hydrocarbon chains, PEG, are used for coating to improve GNP biocompatibility and, meanwhile, prevent the formation of clusters. PEGylation can be achieved by replacing the pristine capping agent of nanoparticles with thiol-terminated PEG [42]. Such ligand exchange increases the cellular uptake of GNPs because of the chemical compatibility of PEG toward cellular membranes [43,44]. PEGylation influences pharmacokinetics and cellular uptake dynamics, reducing GNP clearance, cytotoxicity, and immunogenicity, as well as improving GNP plasma circulation time and tumor cell targeting potential [45,46]. In vitro and in vivo studies carried out on breast, liver, prostate, and ovarian cancer showed that PEGylated glucose-coated GNPs are more effective in targeting solid cancer cells, significantly more than surrounding normal cells [47]. Folic acid, a water-soluble vitamin, and a non-immunogenic agent are employed as active targeting molecules and are extensively investigated for the identification of tumor cells [48]. Indeed, the expression of folate receptors in cancer cells in the ovarian, fallopian tube, kidney, lung, and thyroid cancerous tissues outperforms the folate receptor expression of healthy cells. Moreover, epidermal growth factor (EGF) and suitable monoclonal antibodies are regarded as effective targeting agents for the surface modification of GNPs, as they allow the specific recognition of tumor cells [26,49,50,51]. Some relevant examples of GNPs functionalized with targeting agents and used for tumor ablation by PTT are overviewed in the following section.

## 5. GNPs for Radiotherapy

GNRs offer a versatile framework for incorporating multiple therapeutic modalities. For example, Xu et al. [52] demonstrated chemo-PPTT using doxorubicin (DOX)-GNR complexes. GNRs have also been successfully used in cancer diagnosis bioimaging techniques, such as computed tomography, photoacoustic imaging, and optical coherence tomographic imaging. The radio-sensitizing ability under X-ray irradiation given by a high atomic number (high-Z) of gold gives the possibility to use GNRs for radiotherapy (RT). High-Z enhances X-ray absorption in local tissues, releasing low-energy electrons and consequently more free radicals capable of damaging DNA [53]. Combining RT with PTT, it could be possible to reduce the radiation dose and enhance anti-cancer efficacy: hyperthermia increases the local cancer tissue, softens blood vessels, increases blood circulation and oxygen transport, boosts the sensitivity of hypoxic cancer cells to radiation, and inhibits the mechanism of repair [54]. An example is the study of Sun et al. [55] in which they developed cancer cell membrane-coated gold nanorods (GNR@Mem) for PTT and RT of oral squamous cancer, as well as gold nano-sesame-beads (GNSbs), a gold-nanorod-seeded mesoporous silica nanoparticle [56]. Arginine-glycine-aspartate (RGD)-conjugated GNRs with a layer of silica (RGD-GNRs) and RGD- and ACPP-coated core-shell Au@Se nanocomposites were used to enhance the RT of melanoma cells [57,58], while RGD-conjugated and mesoporous silica-coated GNR (GNRs@mSiO2-RGD) multifunctional nanoprobes were used for RT in MDA-MB-231 triple-negative breast cancer (TNBC) cells [59]. Goserelin-conjugated GNRs [60] and goserelin-conjugated GNRs using erbium (Er)-filtered X-rays were developed for radiosensitization of prostate cancer [61]. Folic acid-conjugated silica-coated GNRs (GNR-SiO2-FA) were realized to selectively target MGC803 gastric cancer cells [62]. Other silica-coated GNRs, folic acid-conjugated silica-coated gold nanorods (GNRs@SiO2-FA), were proposed for the radio sensitization of the hepatocellular carcinoma cell line HepG2 [63]. GNRs loaded with siRNA against the anti-apoptotic sphingo-sine kinase (SphK1) gene were utilized to induce radiation sensitization in patients with head and neck squamous cell cancer (HNSCC) [64].

## 6. ^18^F-FDG: Synergistic Targeting Agent and PET Imaging Nanoprobe

Fluorodeoxyglucose has been utilized successfully in cancer imaging by taking advantage of cancer cells’ glucose avidity. Because cancer cells reproduce quickly, they utilize glucose at a greater rate compared to healthy cells. The Warburg effect is the name given to this phenomenon [65]. The glucose analogue 2-(^18^F)fluoro-2-deoxy-D-glucose (^18^F-2-FDG) is a well-known positron emission tomography (PET) imaging radiotracer (Figure 1).

The most extensively documented metabolic inhibitor for targeting glucose metabolism is the glucose analogue 2-deoxy-D-glucose (2DG), which inhibits glycolytic ATP generation and glucose transport. Furthermore, 2DG can cause oxidative stress, block N-linked glycosylation, and cause apoptosis through endoplasmic reticulum stress [66]. Some earlier studies described the Warburg effect and discussed the mechanisms underlying 2DG and its potential application in cancer treatment [67,68]. Thus, glucose labeling could be an effective method of facilitating GNP internalization into cells. Recently, 2-deoxy-D-glucose modified PEG-coated nanomaterials were created as a possible dual-targeted drug delivery strategy for boosting drug accumulation in glioma via GLUT-mediated endocytosis and enhancing blood-brain barrier passage by Glut-mediated transcytosis [69]. Suvarna et al. described the synthesis of various capped GNPs, as well as the characterization of the nanoparticles, using various approaches. This study was designed to evaluate the potential biomedical application of gold nanomaterials capped with 2DG and citrate using different reducing agents, portraying 2DG-capped GNPs as better candidates for theranostic application [70]. Unak et al. [71] provided a novel architecture by conjugation of ^18^F-FDG, GNPs, and anti-metadherin (anti-MTDH) antibody (see Figure 2). This antibody is specific for MTDH, which is a surface protein overexpressed in breast cancer cells. In detail, this paper proposes a route for the preparation of a nano-bioconjugate that merges the radiopharmaceutical with GNPs and the antibody as an active-targeting agent. The preparation strategy consists first of the synthesis of ^18^F-FDG functionalized through a mannose triflate molecule as a precursor for ^18^F-FDG by nucleophilic fluorination. The obtained radiopharmaceutical carries a thiol group introduced using cysteamine in the synthesis step. The thiol group promotes the in situ generation of GNRs by the reduction of HAuCl4. The final step of the preparation procedure is the bioconjugation of ^18^F-FDG functionalized GNPs by generating a covalent bond between the ^18^F-FDG and the amine groups of the antibody. Roa et al. [72] synthesized a multi-functional, radiosensitizing agent based on thiol-6-fluoro-6-deoxy-d-glucose GNPs (thiol-6-FDG–GNPs). Remarkably, the group of Roa et al. was the first to report the biodistribution, pharmacokinetic evaluation, and toxicological safety of 6-FDG–GNPs both in vitro and in vivo. Moreover, thiol-6-FDG–GNPs were developed with the capability of being labeled by the radioisotope ^18^F for eventual PET imaging. In comparison with ^18^F-2-FDG which is poorly absorbed by the intestines and kidneys, the analogue ^18^F-labeled 6-FDG (^18^F-6-FDG) is actively transported by the kidneys and intestines. After 2 h of intravenous injection of 6-FDG-GNPs into the murine model, approximately 30% of GNPs were detected in the liver, spleen, and kidney. PEGylation of 6-FDG-GNPs was reported to drastically ameliorate 6-FDG-GNP biodistribution by minimizing inadvertent uptake into these organs, while simultaneously tripling the cellular uptake of gold nanoparticle GNPs in implanted breast MCF-7 cancer. This research intends to replace the ^18^F atom in the 6-FDG molecule in the future so that it can be used as a PET radiotracer to facilitate cancer detection and image-guided radiation therapy planning. Increased uptake of Glu-GNPs by cancer cells compared to unbound GNPs leads to improved radiotherapeutic cytotoxicity in vitro [72] and contrast enhancement of tumors in computed tomography (CT) imaging [73]. Some investigators have outlined how gold nanomaterials show significantly higher X-ray absorption than iodinated CT agents, thus providing the opportunity of using GNPs to improve the resolution of functional CT and molecular imaging until the micrometer scale. Feng et al. [74] discussed how they employed FDG-coated GNPs as multimodality CT and PET imaging contrast agents. They demonstrated in vitro and in vivo that cancer cells consume PEG-Glu-GNPs at 10 to 100 times higher concentrations than healthy cells and verified that this nano-compound has much better targeting ability than gold nanoparticles without surface modification. By analogy, Hu et al. [47] investigated whether there is any significant difference in targeted treatment due to glucose conjugation, again in GNPs designed as radiosensitizers; they found that Glu-GNPs are better radiosensitizers and enhance the cancer-killing of specific cancer cells 20% more than X-ray irradiation alone. In Table 1, a summary of the main studies is reported.

## 7. GNP-Mediated PTT for Effective Cancer Ablation

Traditional hyperthermia procedures (photodynamic therapy, radiofrequency hyperthermia, microwave hyperthermia) are still not optimum, since they are not minimally invasive and result in non-specific heat release across the body, which causes unpleasant side effects [75,76]. However, the ability to finely modulate heating around the tumoral zone, which is achieved by photothermal therapy, is highly sought in order to safely optimize hyperthermia. Photothermal therapy is one of the most effective adjuvants to radiotherapy and is a promising approach to eradicate radioresistant cancer cells. Indeed, hyperthermia and PTT, in addition to their direct toxic effect on tumors, can destroy radioresistant malignant cells, such as cancerous cells in hypoxic, low pH areas, and cells in the S-phase, which are primarily responsible for cancer recurrence and malignant dissemination after radiation RT [77,78]. It is worth pointing out that, from a practical point of view, light is an ideal external stimulus, as it can be easily regulated, focused, and remotely controlled to provide better-targeted treatments that lead to less damage to healthy tissues [79,80].

### 7.1. GNPs for Thermoablation of Cancer Cells

Several investigations have established the efficiency of plasmonic GNPs for the thermal ablation of different cellular types by PTT, as summarized by outstanding reviews [7,81,82,83]. Pitsillides et al. established the usefulness of gold nanoparticle GNPs for thermally induced cellular death: anti-CD8-labeled GNPs were used for the selective targeting and killing of T-cells. However, as discussed in the previous section, an efficient in vivo application of GNP-mediated PTT requires GNPs with plasmon bands lying in the “biological window [84]. The coupled LSPR of nanospheres can be exploited by employing aggregated or assembled systems, as suggested by El-Sayed et al. [51]. According to the study, 30 nm gold nanospheres coated with anti-EGFR may be assembled on cell membranes with upregulated EGFR at high concentrations. When compared to the initial state, the formed nanospheres’ absorption signal was redshifted. (non-assembled nanoparticles). As mentioned in Section 3, anisotropic nanoparticles, such as GNRs [85] or nanobypyramids [86] can absorb light in the biological water window, which is extremely advantageous for PTT-related applications. Zhang et al. investigated in vitro NIR plasmonic photothermal therapy using anti-epidermal growth factor receptor (EFGR)-conjugated GNRs [87]. The GNRs bind selectively to the cytoplasmic membranes of malignant-type cells with EGFR hyperexpression. As a consequence, the malignant cells were killed at roughly half the laser strength required to kill non-malignant cells. Later, the researchers proved the viability of in vivo PTT employing polyethylene glycol (PEG)ylated GNRs, observing a significant size decrease in human squamous carcinoma tumors treated with gold nanorods and photothermal therapy (PTT). By altering the surface of popcorn-shaped GNPs with anti-prostate-specific membrane antigen (PSMA) antibodies, Lu et al. created a targeted PTT agent [30] (Figure 3).

After 30 min of irradiation (785 nm cw laser, 12.5 W/cm^2^), LNCaP human prostate cancer cells were treated with anti-prostate-specific membrane antigen GNPs and displayed substantial mortality. Furthermore, Black et al. created polydopamine (PD)-coated GNRs [88]. Anti-EGFR antibodies were attached to the PD-coated GNRs, and the anti-EGFR-conjugated, PD-coated GNRs were effectively employed for targeted PTT. Gold nanoparticle GNPs with more complex morphologies were also studied and evaluated for tumor eradication through PTT. Hirsch et al., for example, described the GNS-mediated NIR PTT of tumors using magnetic resonance guidance [89]. They incubated human breast cancer cells with PEGylated GNSs and found that the cells became photothermally ablated after NIR laser irradiation. Furthermore, Au et al. [90] demonstrated that gold nanocage GNCs have a photothermal effect on SK-BR-3 breast cancer cells (Figure 4).

Gold nanocages conjugated with anti-HER-2 antibodies showed selective photothermal destruction of breast cancer cells. Gold nanostars conjugated to drugs have been applied in tumor treatment. When administered intravenously to mice, PEGylated gold nanostars accumulated for 48 h, extravasation was demonstrated, and localized photothermal ablation was reported within 10 min of irradiation [91]. Chen et al. conjugated GNSTs with cyclic peptide-RGD and an anticancer drug (DOX) to show the synergistic effects of PTT and chemotherapy [92]. Relative to GNRs and GNCs, PEGylated nanohexapods demonstrated the greatest tumor uptake and photothermal conversion efficiency. Simón et al. used 800 nm Resonant BioPureTM Gold NS, which is a silica core surrounded by a thin gold shell, for the murine subcutaneous colorectal tumor model [93]. They applied two fractionated PTT protocols, one with two laser treatments and one with four laser treatments. Since in previous studies an unspecific temperature increase of ΔT ~10 °C determined by laser attenuation in the dermal layer was observed [94,95], they swabbed into the tumor glycerol to enhance transdermal laser penetration, obtaining a significant reduction of maximum temperatures reached. The results demonstrated no substantial differences in tumor growth and survival between mice receiving single doses and mice treated with fractionated PTT. Only a few animals receiving fractionated PTT showed the complete disappearance of cancer. Recently, doxorubicin-encapsulated iron-gallic acid (FeGA-DOX) nanoparticles (NPs) were linked with agarose hydrogels (AG) for PTT in osteosarcoma [96], providing an innovative therapeutic tool, while a smart deoxyribose nucleic acid nanogel-coated polydopamine nanosphere hybrid was developed for chemo-PTT for oncological applications [97].

Breast cancer is a flourishing field of research for nanomaterials and PTT. Different studies have been conducted. An innovative advance was the application of a sub-10 nm supramolecular nano assembly of aluminum and indocyanine green with lignosulfonate (LS-Al-ICG) for breast cancer PTT in a murine model [98]. Firstly, it was observed the capacity of aggregation in the tumor site confirmed the high photothermal-conversion effect of LS-Al-ICG. Since PTT can elicit ICD through the release of tumor-associated antigens (TAAs) and damage-associated molecular patterns (DAMPs) liberated by dying tumor cells [99,100], dendritic cell maturation and cytotoxic T-cell responses were stimulated; therefore, the activation of immune responses was seen. A considerable inhibition of tumor growth was evident, but the immunomodulation of the microenvironment improved efficacy for distant tumors as well. In fact, when the volume of distant tumors was measured, a significant reduction resulted, implying that an abscopal effect occurred. In addition, while in previous studies it was demonstrated to have limited effects on CD8+T cell populations using aluminum hydroxide and aluminum phosphate [101], providing an aluminum adjuvant with ICG elicited better cellular immunity. Another application of PTT in breast cancer is the encapsulation of 6-mercaptopurine (6MP) with chitosan nanoparticles (CNPs) loaded with gold nanomaterials. In particular, the effect on cellular proliferation of the human breast carcinoma cell line MCF7 was demonstrated, comparing free 6MP, which produces maximum inhibition of 39% at 100 μM, 6MP-CNPs with an IC50 at 9.3 μM, and 6MP-CNPs reaching 8.7 μM. The encapsulation of 6MP guarantees higher anti-proliferation activity, an augmented intracellular uptake, and thus a reduction of side effects [102]. Hyaluronic acid (HA)-Au@SiO2@Au NPs were proposed for the treatment of bladder and prostate cancers in mice. The tumors in mice disappeared at 15 and 18 days without signs of recurrence, highlighting the excellent targeted photothermal ablation and high biocompatibility [103].

GNRs surrounded by a polymeric shell constructed employing the layer-by-layer approach in which doxorubicin (DOXO) was incorporated and linked to HA (PSS/DOXO/PLL/HA-coated AuNRs) were proposed. The compound provoked cell death mainly through apoptosis in HeLa cells, even in the presence of NIR light irradiation [104]. Polypyrrole, cystine dihydrochloride, and hyaluronan nanoparticles loaded were tested in MDA-MB-231 breast cancer cells and breast tumor-bearing mice. Good photothermal effects, in vitro efficiency-induced apoptosis, and the inhibitory effect on cancer growth in mice models showed the great potential of chemo-photothermal therapy combination [105].

### 7.2. Lasers in PTT

The distribution of laser flux is an essential element in producing efficient ablative efficiency. Some characteristics, such as laser time and intensity, have been demonstrated to impact cell damage in PTT [106]. Furthermore, lasers in pulsed mode, particularly those with high pulse energy, are commonly associated with significant heating effects, which can have a wide range of implications. Compared to continuous-wave lasers, a pulsed laser with a brief pulse duration delivers more energy to the tissue. Furthermore, because they transfer heat to fluids, pulsed lasers produce a phase change in the water, resulting in the formation of bubbles in the biological system [107]. GNPs treated with a pulsed laser in the cytoplasm were found to create large bubbles. As a result, using pulsed lasers in the presence of GNPs provides the advantage of lowering the pulse laser intensity while maintaining the same effects. The wavelength of the incident light must be chosen carefully for the PTT. Regarding practical uses, the NIR region (650 nm 900 nm) is typically used because it exhibits negligible absorption and scattering by water, hemoglobin, skin, and other biomolecules of tissue components. NIR light penetrates tissue deeply, allowing it to reach deep within the body.

## 8. Effects of GNPs on the Immune System

The effect of GNPs on the immune system is likely to be dependent on their structure, with one study demonstrating that gold nanoparticle GNPs can induce pro-inflammatory responses depending on their size and other studies demonstrating anti-inflammatory responses [108]. GNPs are generally constructed using cetyltrimethylammonium bromide (CTAB), which is highly toxic and lacks functions for surface modification [109]. The host and cancer tumor biology cooperate in various ways to affect the biological destiny and storage of nanoparticles [110]. They can interact in unforeseen ways with the immune system and the solid tumor microenvironment, eventually, and profoundly affecting performance and tumor response in the setting of immuno-activation [111]. There have been limited studies evaluating nanoparticle destiny and intratumor accumulation across biological models and immune cells or tumor compartments. The presence of immune cells within a well-established solid tumor suggests that immune cells are conducting surveillance and homeostatic roles to aid in the tumor’s growth and maintenance. Our findings suggest that nanoparticle exposure can break up this delicate equilibrium, perhaps allowing for transitory immune detection of the tumor. In an immune-compromised cancer model, the systemic distribution of a nanoparticle design can trigger a complicated immune response that can influence tumor growth regardless of nanoparticle retention [112].

## 9. Current Limitations of GNPs for PTT

Although GNPs are generally considered biocompatible and non-toxic, the unique property of gold nanomaterials that could have a significant harmful impact is the uncontrolled generation of reactive oxygen species [113,114]. The eventual GNPs’ biological damages are highly dependent on their physicochemical properties. Positively charged GNPs, for example, have been found to have a stronger inflammatory potential than negatively charged or neutral GNPs [115]. In vitro, Yen et al. demonstrated that GNPs, regardless of size, upregulate the proinflammatory gene expression of interleukin-1, interleukin-6, and tumor necrosis factor (TNF-alpha) [116]. A potential drawback of PTT techniques must be searched in the NIR lasers used that, despite being harmless, are characterized by penetration depth still limited in biological tissues, estimated at 2–3 mm. This low depth of penetration would exclude deeper lesions from the possibility of treatment and would consequently require considering only more superficial tumoral tissues (breast, skin, prostate, selected bone, and lymph nodal regions) as candidates for PTT thermal ablation as safe as effective or combined with an endoscopic procedure in the case of endoscopy-detectable lesions. A further issue that represents one of the main concerns of the biomedical application of GNPs is their long-term cytotoxicity. Surface molecules, such as CTAB, which are essential for the colloidal stability or chemical functionalization of gold nanoparticles, can potentially cause toxicity issues. Because of the physicochemical features of NPs, the long-term toxicity of GNPs due to nonspecific absorption in untargeted organs remains unclear. Further extensive research is needed. Data provided by some authors [117,118,119] showed that a relatively large GNP has a negligible quantity of brain uptake measured. Previous research by De Jong et al. indicated that the tissue location of GNPs is size dependent [117], with the smallest (10 nm) GNPs exhibiting widespread organ dispersion. Hirn et al. found a consistent location of gold nanoparticles (1131 nm, hydrodynamic size) throughout various organs [118]. Neither investigation demonstrates blood-brain barrier penetration by GNP > 10 nm. Guerrero et al. [119] conducted an analogous investigation and found that a moderately large gold nanoparticle template (12 nm Au core) radiolabeled with ^18^F-fluorobenzoate had a similar in vivo distribution to that observed by De Jong et al. and Hirn et al., with a small amount of brain uptake. Nonetheless, other investigations discovered that the time-activity curves of the ^18^F brain distribution derived from PET imaging demonstrated the cerebellar and cerebral uptake of radiolabeled GNPs with time. Unlike bigger nanoparticles (>12 nm), 3 nm NPs continue to accumulate radioactivity in the cerebellum and brain after 120 min. The tunable optical properties of GNRs are indeed well suited for inducing cell death, but these results, although encouraging, have been achieved in non-realistic conditions, namely in experimental conditions far from the physiological state of the targeted cancer cells. Evidence of minimum to mild chronic inflammation was found in the regions around these nanoparticles, which were classified as foreign bodies [120], albeit the long-term implications of this inflammation were not completely understood [121]. While preliminary studies seem promising in terms of potential cytotoxicity, there are still uncertainties about whether GNPs are finally cleared from the body or if probable long-term repercussions result from nanoparticle storage. ^18^F-FDG is expected to favor tumor accumulation. However, even in the most favorable case, the percentage of the injected dose is unlikely to be higher than 10% when gold nanomaterials are administered via intravenous injection. As a matter of fact, to date, GNP studies have been carried out in animal models only and generally for not more than six months of follow-up time [6]; therefore, many questions about GNPs’ long-term body clearance and the eventual effects resulting from gold nanomaterial accumulation in specific organs (i.e., liver, spleen, kidneys) over longer time courses are left unanswered and need for further precise investigations.

## 10. Conclusions

One of the most relevant objectives of current oncological research is succeeding in functional matching between early tumor diagnosis and the precise application of specific effective therapeutic tools. Photothermal plasmonic treatment is a relatively new therapy that might benefit from improved monitoring and evaluation. Even though this treatment has been demonstrated to be beneficial in preclinical studies, the outcome is unpredictable since it is dependent on many biological parameters, including the tumoral distribution of nanoparticles. As a result, the treatment should be personalized, and if no response is obtained, the planning should be modified. Simon et al. established that PET/CT has predictive significance for early PTT therapy evaluation in tumor-bearing mice. Furthermore, ^18^F-FDG has been used in cancer imaging since many cancers have an enhanced rate of glycolysis due to the Warburg effect when compared to healthy tissue [122]. Previous research indicated that a decrease in ^18^F-FDG uptake in the tumor after therapy might be considered tumor viability loss and was associated with improved survival. Furthermore, tumors with low baseline uptake or those located near tissues with high glucose metabolism could suffer from inadequate ^18^F-FDG evaluation. Furthermore, an inflammatory response may be accompanied by increased ^18^F-FDG uptake, which may influence tracer specificity. Because of their great medical potential, these NPs have piqued the curiosity of the molecular imaging and radiology society. The careful production and testing of these NPs probes utilizing nuclear imaging technologies provides significant insight into their in vivo destiny and potential utility in clinical trials and research. These findings show that size, charge, shape, and surface characteristics are important factors for GNPs’ biocompatibility, safety, and clearance. Few nano substances have made it to the evaluation phase in humans, although several nanoplatforms are in the translation planning stages. In a variety of pre-clinical scenarios, radiolabeled nanoparticles have been studied for SPECT and PET imaging in addition to multimodality MR and optical imaging. Our group has recently developed ^99m^Tc-radiolabeled GNPs, resulting in a theragnostic nanoformulation able to merge the photothermal properties of GNPs with the unique properties of ^99m^Tc-DTPA complex to be used for SPECT imaging. GNPs were modified with keratin molecules to provide biocompatibility and were then radiolabeled by exploiting their affinity with a metal chelator (DTPA). Our proof-of-concept was implemented on a PDMS microfluidic device that mimics the structure and composition of the kidney’s filtering unit (the nephron). The investigation pointed out the dynamic stability of ^99m^Tc-DTPA-Ker-AuNPs, the preservation of the photothermal properties of gold nanoparticles, and the suitability of the ^99m^Tc-DTPA-Ker -GNP as a radiotracer for the SPECT imaging technique. Our study pays the way for the development of new theragnostic strategies potentially able to perform multimodal tumor treatments [123]. Indeed, further studies and integrated strategies for cancer treatment by merging PTT ablation with chemotherapy-selective drug delivery or PTT for radiotherapy sensitization purposes are needed. These accomplishments may open new paths and developments for the goal of facilitating effective, safe, and precise cancer treatment modalities.

## Figures and Tables

**Figure 1 pharmaceutics-15-00319-f001:**
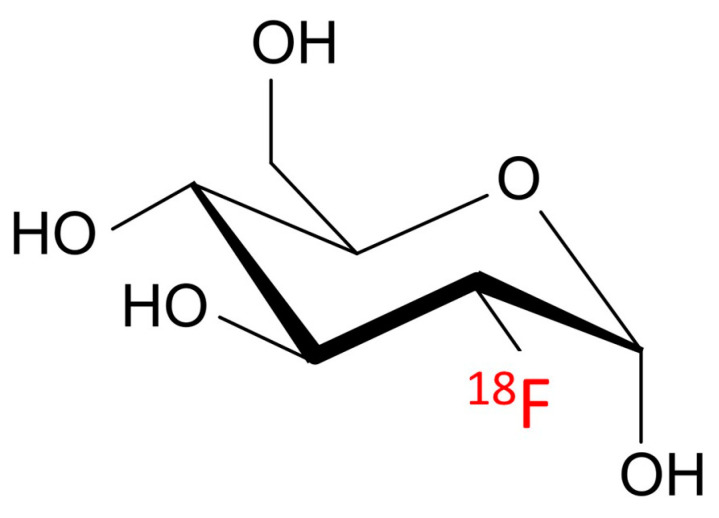
Chemical structure of the glucose analogue 2-(^18^F)fluoro-2-deoxy-D-glucose.

**Figure 2 pharmaceutics-15-00319-f002:**
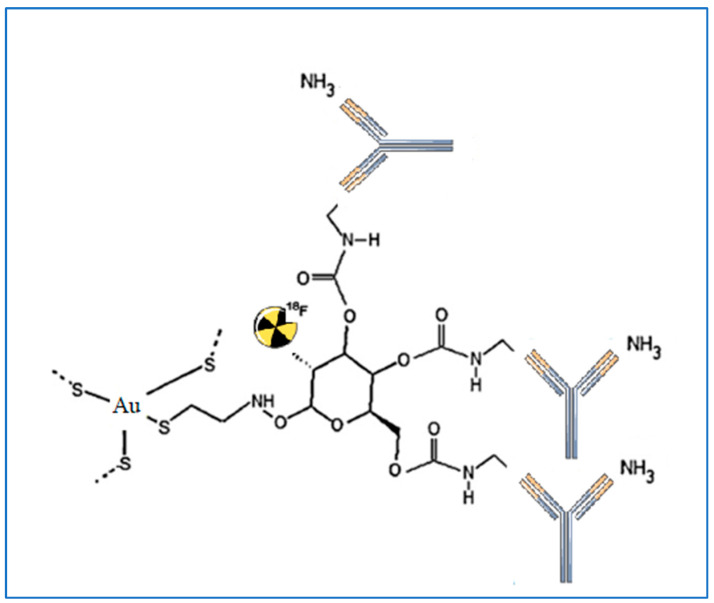
An example of GNP conjugated with the anti-MTDH antibody and labeled with ^18^F-FDG described in the study of Unak et al. [71].

**Figure 3 pharmaceutics-15-00319-f003:**
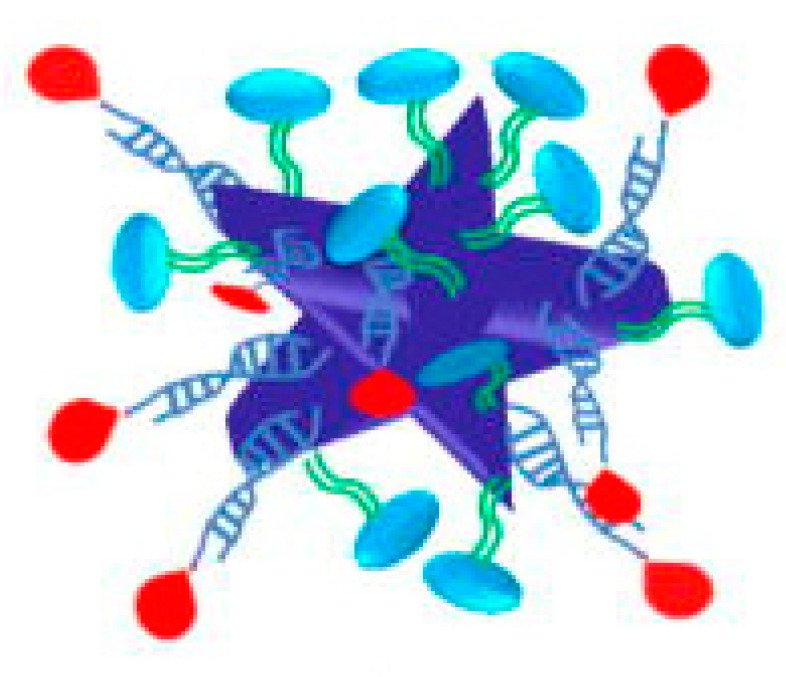
Popcorn-shaped GNPs with anti-prostate-specific membrane antigen (PSMA) antibodies taken by Lu et al. [30].

**Figure 4 pharmaceutics-15-00319-f004:**
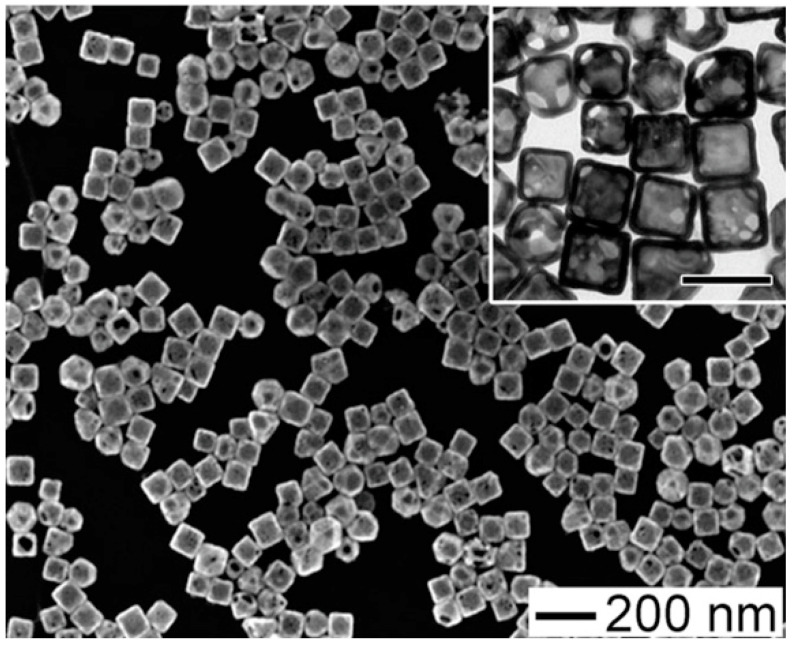
Gold nanocages developed by Au et al. [90] for PPT on SK-BR-3 breast cancer.

**Table 1 pharmaceutics-15-00319-t001:** GNPs modified with FDG-related molecules for biomedical applications.

Author	GNPs Surface Agent	Target	Biomedical Application
Roa et al. (2012) [72]	6-fluoro-6-deoxy-D-glucose	Breast adenocarcinoma	Radiosensitizer
Suvarna et al. (2017) [70]	2-deoxy-D-glucose	cancer cell lines such as HepG2, HeLa and HCT 116	Theragnostic
Hu et al. (2015) [47]	Glucose-coating	Breast adenocarcinoma	Radiosensitizer
Jiang et al. (2014) [69]	PEGylation + 2-deoxy-D-glucose	Brain glioma	Drug-delivery
Unak et al. (2012) [71]	^18^F-2-fluoro-2-deoxy-D-glucose + Ab anti-metadherin (MTDH)	Breast adenocarcinoma	Theragnostic
Feng et al. (2014) [74]	PEGylation + glucose-coating	BALB/c nude mice	CT imaging contrast agent
Hu et al. (2015) [47]	Glucose-coating	Leukemic stem cell line THP-1 and breast cell line MCF-7	Radiosensitizer

## Data Availability

Not applicable.

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
