# Peer review of "^18^F-fluorodeoxyglucose (^18^F-FDG) Functionalized Gold Nanoparticles (GNPs) for Plasmonic Photothermal Ablation of Cancer: A Review"

_pharmaceutics, 2023, doi:10.3390/pharmaceutics15020319_

Round 1

Reviewer 1 Report

This review article reported by Mariano et.al. will give very nice view of possible future applications and limitations of radio labelled Gold Nano Particles for the use of PPT/PTT. This review article contains lot of information on GNPs modified with FDG-related molecules for biomedical applications. I would recommend accepting this publication after answering the following questions.

1.       PPT abbreviation has not given while using it for the first time in line 288.

2.       Chemical structure of 2-(18F)fluoro-2-deoxy-D-glucose is missing.

3.       This review is only having one chemdraw figure that makes readers exhausted reading only the text. Authors left it to the reader’s imagination. It would make readers life easier when they read review along with some example figures of Gold Nano-particles previously reported in the literature.

Author Response

Thank you for the comment 1, the PTT abbreviation has been explicated with the term “Plasmonic photothermal therapy” and better explained in line 113.

Thank you for the comment 2, the chemical structure (yet explicated in the text in line 289) has been added as figure 1.

Thank you for the suggestion. More figures have been added.

Author Response

I would thank you for the suggestion, the term “a review” has been added in the end of the title.

I would thank you for the second advice, all the abbreviations have been identified.

Thank you for the comment 3, the aim has been better clarified.

Thank you for the comment 4, the acronyms have been explicated.

I would thank you for the last observation. The conclusion was represented by the last paragraph, in particular the last lines 605-609 “Indeed, further studies…. treatment modalities.“. Actually it was not clearly expressed. For a better clarification, the title has been changed in “10. Future perspectives and disclosure “
